# Quality of life among patients with cancer and their family caregivers in the Sub-Saharan region: A systematic review of quantitative studies

**Yousef Qan'ir**[1], **Ting Guan**[2], **Eno Idiagbonya**[1], **Cloie Dobias**[1], **Jamie L. Conklin**[3], **Chifundo Colleta Zimba**[4], **Agatha Bula**[4], **Wongani Jumbo**[4], **Kondwani Wella**[5], **Patrick Mapulanga**[5], **Samuel Bingo**[6], **Evelyn Chilemba**[5], **Jennifer Haley**[1], **Nilda Peragallo Montano**[1], **Ashley Leak Bryant**[1], **Lixin Song**[1,7]*

1 School of Nursing, University of North Carolina (UNC), Chapel Hill, NC, United States of America, 2 School of Social Work, The University of North Carolina at Chapel Hill, Chapel Hill, NC, United States of America, 3 Health Sciences Library, The University of North Carolina at Chapel Hill, Chapel Hill, NC, United States of America, 4 University of North Carolina (UNC) Project, Lilongwe, Malawi, 5 Kamuzu University of Health Sciences, Lilongwe, Malawi, 6 Kamuzu Central Hospital, Lilongwe, Malawi, 7 Lineberger Comprehensive Cancer Center, UNC, Chapel Hill, NC, United States of America

* lsong@unc.edu

**Data Availability Statement:** This is a review of published research reports. All studies reviewed are publicly available in the following databases:

## Abstract

Guided by the World Health Organization quality of life (WHOQOL) framework, this systematic review aimed to examine evidence about the prevalence and severity of QOL-related health problems and their influencing factors in Sub-Saharan Africa (SSA). We identified eligible publications in English language from PubMed, Cumulative Index of Nursing and Allied Health Literature Plus with Full Text, Embase, APA PsycInfo, Scopus, and African Index Medicus databases. We included quantitative descriptive studies that measured overall and subdomains of QOL as the outcome in adult patients/survivors with cancer in SSA. Twenty-six descriptive cross-sectional studies (27 papers) that were conducted since 1988 in different SSA countries among patients with various types of cancer met our inclusion criteria. We found inconsistencies in how the prevalence and severity of QOL-related health problems have been researched and reported across studies, which complicated comparing findings and drawing conclusions. The most common factors that influenced the overall and subdomains of QOL included coping; internal and external locus of control; symptoms and symptom management; and religious beliefs and religious care. Demographics (e.g., age and marital status), cancer-related factors (cancer stage and type of treatment), and social determinants of health (e.g., education, access to information and resources, financial distress, and urban vs rural residency) also impacted QOL and its subdomains. Our findings indicate the significant need for recognizing and managing QOL-related problems for cancer patients and caregivers in SSA. Research needs to use culturally adapted, standardized assessment tools and analysis approaches to better understand the QOL challenges this population faces. Comprehensive supportive care is needed to address the complex QOL issues in resource-limited SSA.

PubMed, CINAHL (EBSCOhost), Embase (Elsevier), APA PsycInfo (EBSCOhost), Scopus (Elsevier), and African Index Medicus.

**Funding:** The work of Song has been partially supported by R01NR016990 National Institute of Nursing Research (PI: LS). The work of YQ and TG has been supported by the University Cancer Research Fund (UCRF), Lineberger Comprehensive Cancer Center at the UNC-CH (PI: LS). The funders had no role in study design, data collection and analysis, decision to publish, or preparation of the manuscript.

**Competing interests:** The authors have declared that no competing interests exist.

## Introduction

Cancer is becoming a major health issue in Sub-Saharan Africa (SSA) [1]. An estimated 752,000 new patients were diagnosed with cancer in 2018 [2, 3], and the number of new cancer cases per year in SSA is projected to increase by 70% between 2012 and 2030 [4]. These cancer patients' care needs are not being met because of competing demands such as high prevalence of communicable diseases and limited healthcare resources [5, 6], which contribute substantially to the high mortality rate and poor quality of life (QOL) among cancer patients [7]. While QOL is one of the important health outcomes used to evaluate healthcare quality and survivorship experiences among patients with cancer in many parts of the world, we have limited knowledge about QOL-related experiences among cancer patients and their families in SSA. Additionally, the unique sociocultural context (e.g., values and healthcare systems) may be vastly different from the western world where QOL has been extensively researched.

The World Health Organization (WHO) has defined QOL as "individuals' perceptions of their position in life in the context of the culture and value systems in which they live and in relation to their goals, expectations, standards and concerns [8]." Cancer patients' perceived QOL is strongly impacted by their perception of their living conditions and their level of satisfaction with them. To understand the complex concept of QOL, WHO proposed a comprehensive, multidimensional framework (WHOQOL) that depicts QOL as a composite of subdomains of physical and psychological health, level of independence, social relationships, spiritual and religious beliefs, and environmental features [8].

To better understand the QOL of cancer patients related to the SSA sociocultural values and context when they cope with their diagnosis and treatment, this review used the WHOQOL framework to guide our systematic synthesis of evidence about the prevalence and severity of overall and subdomains of QOL and their influencing factors. Our findings will inform development of culturally sensitive supportive care in oncology practice appropriate for SSA [9].

## Methods

We used the 2020 Preferred Reporting Items for Systematic Reviews and Meta-Analyses (PRISMA) to guide the review process (S1 Checklist) [10]. This review was registered on PROSPERO, the International Prospective Register of Systematic Reviews (Registration ID: CRD42020152838) (https://www.crd.york.ac.uk/prospero/display_record.php?RecordID=152838)).

### Eligibility criteria

Studies were included if they: 1) were conducted among adult cancer patients in SSA; 2) utilized quantitative descriptive research methods; 3) measured and reported QOL outcomes (overall or subdomains); and 4) were published as full-text articles in English.

### Information sources and search strategy

A health sciences librarian searched from the dates of inception through the final search date of June 7, 2021, in the following databases: PubMed, CINAHL (EBSCO*host*), Embase (Elsevier), APA PsycInfo (EBSCO*host*), Scopus (Elsevier), and African Index Medicus. The search included a combination of keywords and subject headings related to the research aim: psychosocial or supportive care, Sub-Saharan Africa, and cancer. Conference abstracts were removed from the Embase search to match our inclusion criterion of full-text articles. No other search limits, including date limit, were applied to our search. The PubMed search strategy was developed first and then adapted for the other databases. The complete, reproducible search strategy

for all databases is available in (S1 Appendix). We also searched the African Journals Online (AJO) database, but our search returned no new relevant articles.

## Study selection

All results were exported to Endnote X8 (Philadelphia, Pennsylvania, USA) where duplicates were removed. The remaining studies were placed into Covidence systematic review software (Veritas Health Innovation, Melbourne, Australia, available at www.covidence.org) to organize and complete the screening process. Each title and abstract were first screened independently by two researchers for the eligibility criteria. Each full-text article was then assessed for eligibility by two researchers independently. Group discussions resolved conflicts at both stages.

## Quality assessment

The quality of the publications was assessed using the 2018 version of the Mixed Methods Appraisal Tool (MMAT) [11], a tool that evaluates the methodological quality of studies using qualitative, quantitative, and mixed methods. There are five methodological quality criteria, and each criterion was evaluated as yes, no, or could not determine responses. A detailed report of the rating of each criterion was used to inform the quality of the included studies, rather than calculating an overall score from the ratings of each criterion [11]. Two researchers independently assessed the quality of the studies. Disagreement was resolved by discussion between the two reviewers. When this discussion did not resolve the disagreement, a third reviewer was consulted.

## Data extraction and synthesis

Guided by the WHOQOL framework that consists of a multi-dimensional profile of domains and subdomains (facets) of QOL (Table 1) [8], members of our research team used Excel to independently extract and report data including physical health, psychological health, level of independence, social relations, environment, and spirituality/religious beliefs. We resolved discrepancies through ongoing discussion. A meta-analysis of the outcomes was deemed impossible due to incomplete and heterogeneous information reported in these studies (e.g., populations, measurements, and outcomes). We conducted narrative analysis to synthesize our findings.

# Results

## Study characteristics

The initial search yielded 2,071 publications (Fig 1). After removal of duplicate papers and ineligible studies, 27 articles met our inclusion criteria (N = 26 studies: two papers reported different outcomes of the same study) [12–38].

Characteristics of the included studies are provided in Table 2. These studies used descriptive prospective (n = 25) and descriptive retrospective (n = 1) cross-sectional designs. The studies were conducted in Nigeria (n = 9), South Africa (n = 6), Kenya (n = 5), Ghana (n = 2), Uganda (n = 1), and Ethiopia (n = 2). One study was conducted in both South Africa and Uganda. Only three studies were guided by theoretical/conceptual frameworks, and they included the 'Theory of Positive Psychology' [13], the 'Need Theory by Virginia Henderson' [29], and the 'Resiliency Model of Stress, Adjustment and Adaptation' [19].

## Participant characteristics

The study sample sizes ranged from 21 to 429 patients (total 3,755 patients; mean: 139). Three studies included both cancer patients and their caregivers, and the others focused only on

**Table 1. Structure of the World Health Organization (WHO) QOL domains and facets.**

| Domain | Facet |
|---|---|
| **Physical Health** | Pain and discomfort |
| | Energy and fatigue |
| | Sleep and rest |
| **Psychological Health** | Positive feelings |
| | Thinking, learning, memory and concentration |
| | Self-esteem |
| | Bodily image and appearance |
| | Negative feelings |
| **Level of Independence** | Mobility |
| | Activities of daily living |
| | Dependence on medication or treatments |
| | Work capacity |
| **Social Relationships** | Personal relationships |
| | Social support |
| | Sexual activity |
| **Environmental Health** | Physical safety and security |
| | Home environment |
| | Financial resources |
| | Health and social care: accessibility and quality |
| | Opportunities for acquiring new information and skills |
| | Participation in and opportunities for recreation/ leisure activities |
| | Physical environment (pollution/noise/traffic/climate) |
| | Transport |
| **Spirituality and Religious Beliefs** | |

patients (n = 22) or caregivers (n = 1). These studies included patients with different types of cancer (n = 12) or those with one type of cancer such as breast (n = 9), prostate (n = 2), cervical (n = 2), and head/neck cancer (n = 1). Regarding the stage of cancer, 22 studies included patients with cancers at various stages, whereas four studies recruited patients with advanced cancers.

The mean age of patients across all studies was 49.5 years (SD = 7.1; range = 29–68; median = 49.2). Twelve studies included only females and one study included only men with prostate cancer. Among the 13 studies that included patients of both genders, ten predominantly focused on females and one study failed to specify participants' gender.

## QOL measurements

A wide variety of QOL assessment measurements were used in these 27 articles. Fourteen previously developed measurement surveys were used in different studies, seven studies used self-reported surveys adapted from different questionnaires, and three used researcher-developed surveys. Some studies combined more than one survey type in their assessments. Table 3 identifies the QOL measurements used in each study.

## WHOQOL-related findings

**Overall QOL.** Thirteen studies reported overall QOL findings among patients (Table 3). Among these studies, 14%-56.7% of participants in three studies reported that their QOL

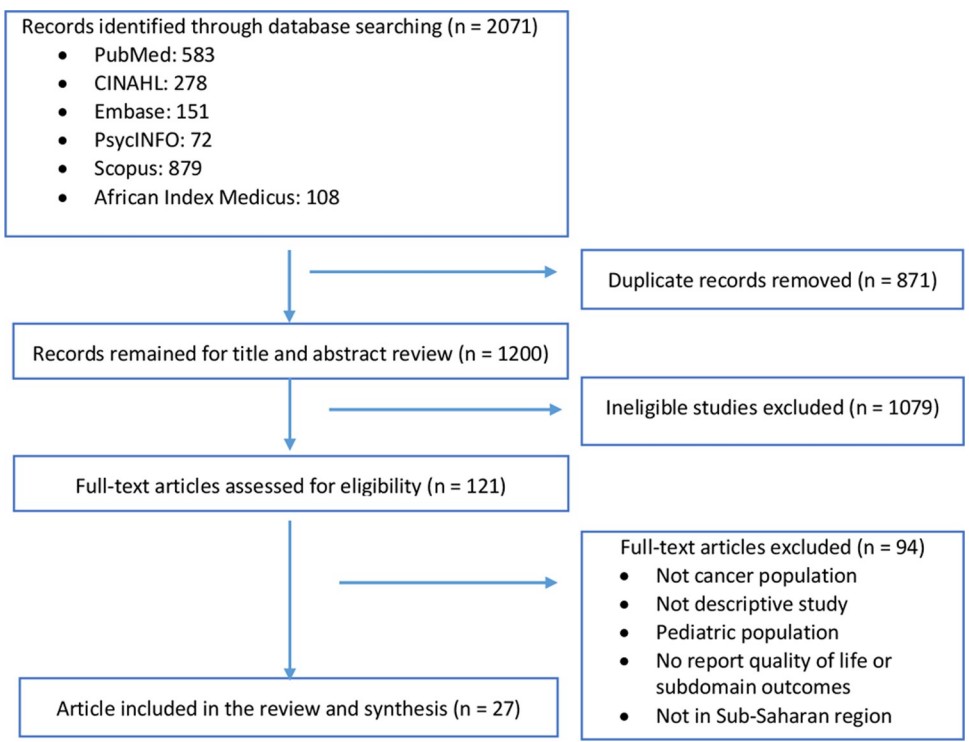

**Fig 1. PRISMA flowchart of study identification and selection.**

varied between fair and excellent. Seven studies reported that the mean scores of overall QOL among patients were low (n = 5), moderate (n = 1), or high (n = 1). Only one study reported that the mean score of overall QOL among caregivers was moderate. One of two studies reported that 53.2% of caregivers had high QOL.

Seven studies examined the factors that influenced overall QOL. Specifically, overall QOL was positively correlated with active coping, religious coping, acceptance, health literacy, and access to health information (n = 3), and negatively correlated with depression and anxiety (n = 2), psychosocial needs (n = 1), and difficulties in health finance (n = 1). In addition, better overall QOL was related to being younger, female, married, in an initial cancer stage, having a higher level of education, and receiving a combination of chemotherapy and radiotherapy (n = 3).

**Physical health.**   Nine studies reported the prevalence of physical health-related problems. In these studies, 16.0%-87.5% of the participants reported pain (n = 7); 71.8%-78.0% reported fatigue (n = 3); 75.0% reported uncontrolled symptoms (n = 1); 42%-80.0% reported sleeping difficulties (n = 7); 80.5% reported weight loss (n = 1); and 15.4% reported motor deficit (n = 1).

Ten studies reported the mean scores of the physical health domain of QOL among patients or caregivers. Two studies reported favorable mean scores of physical health including high level of physical wellbeing (n = 1) and low level of cancer-related symptoms (n = 1). In contrast, seven studies reported unfavorable mean scores of physical health including great (n = 2) or moderate (n = 1) cancer-related symptoms; low level of physical wellbeing (n = 4); high (n = 2) and moderate (n = 1) level of pain; low level of energy (n = 1); weight loss (n = 1); and great sleeping difficulties (n = 2).

**Table 2. Characteristics of study and participants.**

| First Author, Year, & Country research conducted | Study Aim | Sample Size | Cancer Type and Stage of Survivorship | Mean Age (Years) | Gender |
|---|---|---|---|---|---|
| Abebe, 2020 Ethiopia | Assess mastectomy-related QOL | 86 PT | Breast cancer post mastectomy | PT:43.23 | PT: 100% Female |
| Brown, 2012 South Africa | Investigate emotional intelligence and locus of control | 67 PT | Breast cancer receiving treatment | PT 40–59: 64% | PT: 100% Female |
| Cooper, 2001 South Africa | Explore the effect of culture on health-related QOL among cancer patients of extremely low socio-economic status | 167 PT | Breast, lung, esophageal, ovarian and hepatic receiving treatment | PT: 47.3 | PT: 71% Female |
| Elumelu, 2015 Nigeria | Assess the impact of active coping, religion and acceptance on the QOL | 110 PT | Breast cancer receiving treatment | PT: 47.04 | PT: 96% Female |
| Esan, 2020 Nigeria | Determine the coping strategies employed by cancer patients and their effects on QOL | 90 PT | Mixed cancer receiving treatment | PT: 29 | PT: 66% Male |
| Fatiregun, 2017 Nigeria | Evaluate association between anxiety disorders and QOL | 200 PT | Breast cancer | PT:49.6 | PT: 100% Female |
| Gabriel, 2021 Nigeria | Examine the association of needs, health literacy, and QOL among cancer patients and family caregivers | 120 dyads | Mixed cancer in treatment, stage 2 to 4 | PT:62.22 | PT: 70.8% Female; |
| | | | | CG:36.13 | CG: 83.3% Female |
| Greeff, 2012 South Africa | Identify QOL aspects and their association with the successful adaptation of the family caregivers of prostate cancer patients | 21 dyads | Prostate cancer at least 6 months since diagnosis | PT:68 CG:64 | PT: 100% Male CG: 100% Female |
| Harding, 2011 South Africa and Uganda | Determine symptom prevalence and burden | 112 PT | Advanced mixed cancer | PT: 56.6 | PT: 64% Female |
| Ibrahim 2019 Nigeria | Retrospectively assessed QOL | 52 PT | Advanced breast cancer with brain metastasis | PT: 44.7 | PT: 100% Female |
| Kamau 2007 Kenya | Assess QOL of women diagnosed with inoperable cervical carcinoma | 152 PT | Cancer of cervix received radiotherapy | PT: 50–59 | PT: 100% Female |
| Kizza, 2020 Uganda. | Explore the influencing factors of QOL among family caregivers of adult cancer patients | 284 PT; 284 CG | Mixed cancer at different stages | PT: 50.2; CG: 36 | PT: 63.7% Female; CG: 73.2% Female |
| Kugbey 2019 Ghana* | Examine the direct and indirect effects of depression and anxiety on QOL through social support and religiosity | 205 PT | Breast cancer | 52.49 | PT: 100% Female |
| Kugbey. 2019 Ghana* | Examine the direct and indirect influences of health literacy and access to health information on QOL | | | | |
| Kyei, 2020 Ghana | Assess the impact of demographic and clinical characteristics on QOL | 120 PT | Cervical cancer undergoing radiotherapy | PT: 56.8 | PT: 100% Female |
| Marete, 2010 Kenya | Explore palliative care and symptom management issues | 45 PT | Mixed cancer, and terminally ill with wounds | PT: 49 | PT: 60% Female |
| Ndetei, 2018 Kenya | Examine the psychological well-being and social functioning and their influencing factors | 389 PT | Mixed cancer at different stages | PT: 46 | PT: 73.3% Female |
| Ndiok, 2018 Nigeria | Assess palliative care needs from the perspective of the patients themselves. | 429 PT | Mixed cancer | PT: 44.8 | PT: 61% Female |
| Ogoncho, 2016 Kenya | Examine QOL and the influencing factors | 108 PT | Gynecological cancer receiving palliative care | PT: 49.1 | PT: 100% Female |
| Ohaeri, 1998 Nigeria | Examine psychosocial issues and influencing factors | 106 PT | Cervical and breast cancer undergoing radiotherapy | Cervical cancer: 51.2; Breast cancer: 44.9 | PT: 100% Female |
| O'Hare, 1988 South Africa | Assess and compare the psychosocial experience of women who had undergone mastectomy with women who had benign breast tumors removed | 47 PT | Breast Cancer undergo mastectomy undergone breast biopsies | PT undergo mastectomy: 50; PT with benign breast tumor:46.6 | PT:100% Female |

(*Continued*)

**Table 2.** (Continued)

| First Author, Year, & Country research conducted | Study Aim | Sample Size | Cancer Type and Stage of Survivorship | Mean Age (Years) | Gender |
|---|---|---|---|---|---|
| Okoli, 2019 Nigeria | Examine QOL in breast cancer patients | 60 PT | Locally advanced breast cancer (stage IIIA-C). | PT: 48.5 | PT:100% Female |
| Onyeneho, 2021 Nigeria | Examine burden, physical, psycho-social, and financial outcomes among caregivers of cancer patients | 182 CG | Mixed cancer at different stages | CG:37.68 | CG: 60.4% Female |
| Ratshikana-Moloko 2020 South Africa | Identify religious and spiritual needs and assess associations between receipt of religious and spiritual care and patient QOL | 324 PT | Breast, gastrointestinal, and lung carcinoma, and soft tissue sarcoma receiving palliative care services | PT > 50: 73.8% | PT: 52.4% Female |
| Rayne, 2017 South Africa | Examine the fears associated with breast cancer and the influencing factors | 263 PT | Breast cancer undergoing treatment | PT: median age 52 | PT:100% Female |
| Wang'ombe, 2021 Kenya | Find out nature of recovery outcomes among cancer patients attending palliative care | 96 PT | Mixed cancer at different stages | N/A | N/A |
| Yifru,2021 Ethiopia | Assess swallowing function and its impact on QOL | 102 PT | Head and neck cancer at different stages | PT: 42.58 | PT: 53.9% Male |

* One study with two published reports

** The only study used retrospective design across Observational/quantitative descriptive studies; PT = Patient; CG = Caregiver; QOL = Quality of life; N/A = Not available.

Two studies involving caregivers reported contradicting findings, with one study reporting high caregiver physical wellbeing and the other reporting high physical burden of caregiving and poor eating and sleeping.

Six studies reported factors influencing physical health. Better physical health was positively related to active coping, sufficient finance, higher level of education, being married, and use of a combination of treatments (e.g., chemotherapy and radiotherapy) (n = 3). Receiving religious and spiritual care was negatively correlated with patients' pain level (n = 1), while caregivers' perceived burden was positively correlated with patients' pain level (n = 1). Anxiety level was correlated with fatigue and more sleeping difficulties (n = 1).

**Psychological health.** Seven studies reported the prevalence of psychological health-related problems, where 26.9%-67.8% of the participants reported difficulty concentrating (n = 3), reduced memory (n = 1), and cognitive function change (n = 1); 16.7%-79.0% reported depression (n = 5) or sadness (n = 1); 69.6%-78.0% reported anxiety (n = 3); and 44.2% reported post-traumatic stress disorder (PTSD) (n = 1). Additionally, 77.6% reported not experiencing pleasure (n = 1); 72.3% reported negative body image (n = 1); 73.0% reported negative perspective toward the cancer situation (n = 1); and 3.5%-30.9% reported loss of confidence (n = 2). Seven studies reported the mean scores of psychological health domain. Three studies reported favorable psychological QOL, including high level of emotional wellbeing (n = 1) and emotional intelligence (i.e., participants' ability to perceive, express, and regulate emotions) (n = 1), low level of future perspective (i.e., feeling that something is likely to happen because of cancer) (n = 1), and low level of depression and anxiety (n = 1). However, four studies reported unfavorable psychological health among patients, such as low level of psychological wellbeing (n = 1) and emotional wellbeing (n = 2), and high level of negative body image (n = 1). Two studies reported low psychological wellbeing (n = 1) and low psychological burden of caregiving among caregivers (n = 1).

Ten studies reported factors influencing psychological health. Emotional wellbeing was positively related to active coping, religious coping, higher level of education, and being married (n = 2), but negatively related to difficulties in health finance (n = 1), having mastectomy

**Table 3. WHOQOL related findings.**

| Study | QOL Measurements | Reported Domain | WHOQOL Related Findings |
|---|---|---|---|
| Abebe 2020 | Selected items of EORTC QLQ-C30and EORTC QLQ-BR23. | Overall QOL | * Low overall QOL (mean: 48.25). |
| | | Physical Capacity | * High level of postoperative breast symptoms (mean: 19.1) and arm symptoms (mean: 24.5). |
| | | Psychological | * High level of body image (mean: 69.3); Low level of perspective toward the future (mean: 40.3) |
| | | | # Older age, low level of education, being married, unemployed, and living in urban environment were positive toward body image and future perspective. |
| | | Social Relations | * Low level of sexual functioning (mean: 85.3). |
| | | | # Younger age, being married, and living in urban environment were likely to have satisfactory sexual functioning. |
| Brown 2012 | The Schutte Emotional Intelligence Scale | Psychological | * High level of emotional intelligence (mean: 138.18). |
| | | | # Patients with higher levels of emotional intelligence had more internal locus of control orientations, while patients with lower emotional intelligence had more external locus of control orientations. |
| Cooper 2001 | FACT-G | Physical Capacity | * High level of physical wellbeing (mean: 17.6). |
| | | Psychological | * High level of emotional wellbeing (mean: 17.2). |
| | | Independence | * High level of functional wellbeing (mean: 20.4). |
| | | Social Relations | * High level of Social and family wellbeing (mean: 21.7). |
| Elumelu 2015 | FACT-B | Overall QOL | # Active coping, religious coping and acceptance were positively correlated with overall QOL. |
| | | Physical Capacity | # Active coping was positively corelated with physical wellbeing. |
| | | Psychological | # Active coping and religious coping were positively correlated with emotional wellbeing. |
| | | Independence | # Active coping, religious coping and acceptance were positively correlated with functional well-being |
| | | Social Relations | # Active coping, religious coping and acceptance were positively correlated with social wellbeing. |
| Esan 2020 | Validated semi-structured questionnaire adapted from the survey tools developed by Endler and Parker | Overall QOL | ^ Fair overall QOL (56.7%) and good overall QOL (31.1%). |
| | | Physical Capacity | ^ Pain (51%); sleeping difficulties (63.3%). |
| | | Independence | ^ Difficulties in daily activities (49%). |
| | | Social Relations | ^ Reduced social activities (26.7%). |
| | | Environment | ^ Limitation in leisure activities/hobbies (65.6%). |
| | | Spirituality/Religion | ^ Engagement in religious and spiritual activities (88.9%). |
| Fatiregun 2017 | THE EORTC QLQ-C30 | Physical Capacity | # Anxiety level was correlated with lack of energy and sleeping difficulties. |
| | | Psychological | # Financial difficulties were correlated with anxiety. |
| | | Social Relations | # Anxiety was negatively correlated with social wellbeing. |
| Gabriel, 2021 | City of Hope QOL Patient/Cancer Survivor Version; City of Hope QOL (Family Version) | Overall QOL | * Low overall QOL for patients (mean: 159). |
| | | | * Low overall QOL for caregivers (mean: 180). |
| | | | # Psychosocial needs were negatively associated with QOL for patients and caregivers. |
| | | | # Health literacy was positively associated with QOL for patients and caregivers. |
| | | Physical Capacity | * Low physical wellbeing for patients (mean: 35.22). |
| | | | * High physical wellbeing for caregivers (mean: 26.54). |
| | | Psychological | * Low psychological wellbeing for patients (mean: 63.14). |
| | | | * Low psychological wellbeing for caregivers (mean: 73.78). |
| | | Social Relations | * Low social wellbeing for patients (mean: 33.17). |
| | | | * High social wellbeing for caregivers (mean: 45.42). |
| | | Spirituality/Religion | * Low spiritual wellbeing for patients (mean: 28.21). |
| | | | * High spiritual wellbeing for caregivers (mean: 35.50). |
| Greeff 2012 | Self-reported survey adapted from six questionnaires that measured family adaptation and aspects of family functioning | Social Relations | ^ Family support PT:29%, CG: 38%. |
| | | | # Family adaptation was positively correlated with social support. |
| | | Spirituality/Religion | ^ Engagement in religious and spiritual activities PT: 24%, CG: 5%. |

*(Continued)*

**Table 3.** (Continued)

| Study | QOL Measurements | Reported Domain | WHOQOL Related Findings |
|---|---|---|---|
| Harding 2011 | MSAS-SF | Physical Capacity | ^ Pain (87.5%), lack of energy (77.7%), sleeping difficulties (42.0%). |
| | | Psychological | ^ Anxiety (69.6%), feeling sad (75.9%), difficulty in concentration (42.9%). |
| | | Independence | ^ Difficulties in mobility (75.0%). |
| | | Social Relations | ^ Reduced sexual functioning (38.4%). |
| Ibrahim 2019 | Simple WHO performance status | Physical Capacity | ^ Pain (30.8%) and motor deficit (15.4%). |
| | | Psychological | ^ Reduced cognitive function (26.9%). |
| Kamau 2007 | QOL Structured questionnaire and EORTC QLQ-C30 | Overall QOL | ^ Excellent overall QOL (14%). |
| | | Physical Capacity | ^ Pain (72%), lack of energy (78%), and sleeping difficulties (63%). |
| | | Psychological | ^ Loss of confidence (30.9%), difficulty in concentration (46%), reduced memory (56%), depression (79%), anxiety (78%). |
| | | Independence | ^ Difficulties in daily activities (75%). |
| | | Social Relations | ^ Social and family support (56.6%-67.1). Reduced family functioning (71%) and reduced social activities (63%). Satisfactory sexual functioning (11.2%). |
| | | | # Younger age had higher family support and reported satisfactory sexual functioning. |
| | | Environment | ^ Reduction in income (52.6%), extra health expenses (47.4%), and difficulties in finance (63%) and limitation in leisure activities/hobbies (69%). |
| Kizza 2020 | Katz Index, Family Pain Questionnaire, modified Chronic Pain Self-Efficacy Scale and the Caregiver Quality of Life-Index-Cancer. | Overall QOL | ^ CG: High QOL (53.2%). |
| | | | * CG: Moderate overall QOL (mean: 70.2). |
| | | Physical Capacity | * PT: Moderate pain level (mean: 6.14) and Low level of symptoms (mean: 4.5). |
| | | | # The family caregivers' perceived burden was positively correlated with patients' pain level. |
| | | Independence | *PT: Low functional wellbeing (mean: 4.49). |
| | | | # The family caregivers' perceived burden was negatively correlated with patients' functional wellbeing. |
| Kugbey 2019 | FACT-B | Overall QOL | * High level of overall QOL (mean: 95.4). |
| | | | # Depression and anxiety were negatively correlated with QOL. |
| | | Psychological | * Low levels of depression (mean: 5.92) and anxiety (mean: 7.51). |
| | | Social Relations | * High level of social support (mean: 44.65). |
| | | | # Depression was negatively correlated with social support. |
| | | Spirituality/Religion | * High level of religious and spiritual beliefs (37.2). |
| Kugbey. 2019 | FACT-B | Overall QOL | # Health literacy and access to health information were positively correlated with the overall QOL. Anxiety and depression were negatively correlated with the overall QOL. |
| | | Psychological | # Access to health information was negatively correlated with depression and anxiety. |
| #Kyei 2020 | FACT-G | Overall QOL | ^ Good overall QOL (56%). |
| | | | # Younger age, higher level of education, being married, and combination of chemotherapy and radiotherapy were correlated with better overall QOL. |
| | | Physical Capacity | # Higher level of education, being married, and combination of chemotherapy and radiotherapy were correlated with better physical wellbeing. |
| | | Psychological | # Higher level of education and being married were correlated with better emotional wellbeing. |
| | | Independence | # Higher level of education, being married, and use of combined chemotherapy and radiotherapy were correlated with better functional wellbeing. |
| | | Social Relations | # Higher level of education, being married were correlated with better social wellbeing. |
| Marete 2010 | FACT-G, and The Functional Assessment of Chronic Illness Therapy | Physical Capacity | * Low level of physical wellbeing (mean: 2.39–2.95); high level of pain (mean:2.88); lack of energy (mean: 2.43); and sleeping difficulties (mean:1.54). |
| | | | ^ Sleeping difficulties (80%) and pain (16%). |
| | | Independence | * Low level of ability to work (mean: 0.64). |
| | | Social Relations | * Moderate family support (mean: 3.26). low social support (mean: 2.43) and low level of sexual functioning (mean: 0.95). |
| | | Spirituality/Religion | * Low to high level of religious and spiritual beliefs (1.74–3.21). |

(*Continued*)

**Table 3.** (Continued)

| Study | QOL Measurements | Reported Domain | WHOQOL Related Findings |
|---|---|---|---|
| Ndetei 2018 | Structured questionnaire used different measures to assess different facets of respondent's life and psychological well-being. | Psychological | ^ Depression (16.7%) and PTSD (44.2%). |
| | | Independence | ^ Inability to work (26.8%), and difficulties in daily activities (18.9%) and difficulties in mobility (38.7%). |
| | | | # Advanced cancer stage was positively correlated with inability to work. |
| | | Social Relations | ^ Satisfactory social wellbeing (64.3%). |
| | | | # Health related problems negatively correlated with social wellbeing. |
| ^Ndiok 2018 | Structured self-administered checklist with 'yes' or 'no' options developed from the literature. | Physical Capacity | ^ Pain (80%); Lack of energy (71.8%); sleeping difficulties (62.1%). |
| | | Psychological | ^ Difficulty in concentration (67.8%); depression (77.9%); Not experiencing pleasure (77.6%); negative body image (72.3%) and negative perspective toward the situation (73.0%). |
| | | Independence | ^ Difficulties in doing housework' (69.2%) and in mobility (60.1%). |
| | | Social Relations | ^ Reduced sexual functioning (56.2%), social activities (62.5%) and social support (66.6%). |
| | | Environment | ^ Extra health expenses (77.9%); reduction in income (74.1%); and difficulties in accessing care (66.9%). |
| | | Spirituality/Religion | ^ Difficulties concerning the meaning of death (55.0%). |
| Ogoncho, 2016 | Structured questionnaire adopting the MVQOLI | Overall QOL | * Moderate overall QOL (mean: 17.2). |
| | | Physical Capacity | * Moderate level of symptom (mean:8.2). |
| | | | ^ Symptoms well controlled (75%). |
| | | Social Relations | * Low level of social wellbeing (mean: 5.3). |
| | | | # Pain relief and other symptom management are positively correlated with social wellbeing. |
| Ohaeri 1998 | GHQ-12 –a modified version of the German questionnaire by Sullwold and Goldberg | Overall QOL | # Patients with cervical cancer reported better QOL than patients with breast cancer. |
| | | Physical Capacity | ^ Sleeping difficulties (44.3%). |
| | | Psychological | ^ Depression (27.8%) and loss of confidence (3.5%). |
| | | Independence | ^ Inability to work (30%). |
| O'Hare 1988 | Tennessee Self-concept Scale, and Dyadic Adjustment Scale | Psychological | ^ Depression and anxiety (77%). |
| | | Social Relations | ^ Reduced sexual functioning (19%), reduced social wellbeing (27%), reduced social support (50%). |
| Okoli 2019 | FACT-B | Overall QOL | * Low overall QOL (mean: 53.49). |
| | | Physical Capacity | * High level of breast cancer symptoms (mean: 21.1). Low level of physical wellbeing (mean: 10.95). |
| | | Psychological | * Low level of emotional wellbeing (mean: 6.98). |
| | | | # Having mastectomy and younger age were associated with low emotional wellbeing. |
| | | Independence | * High level of functional wellbeing (mean: 17.15). |
| | | Social Relations | * High level of social and family wellbeing (mean: 18.41). |
| Onyeneho, 2021 | Developed from literature review to measure the perceived outcomes of caregiving, and Zarit Burden Interview (ZBI) Questionnaire | Physical Capacity | * High physical burden of caregiving experienced by caregivers (mean: 2.58). |
| | | | * Poor eating pattern (mean: 2.80). |
| | | | * Sleeping difficulties (mean: 2.92). |
| | | Psychological | * Low psychological burden of caregiving experienced by caregivers (mean: 1.88). |
| | | Independence | * Difficulties in daily activities (mean: 3.12). |
| | | Social Relations | * High social burden of caregiving experienced by caregivers (mean: 2.42). |
| | | | * Reduced social activities (mean: 2.45). |
| | | | * Poor family support (mean: 3.75). |
| | | Environment | * High financial burden of caregiving experienced by caregivers (mean: 2.14). |
| | | | * Difficulties in finance (mean: 2.59). |

(*Continued*)

**Table 3.** (Continued)

| Study | QOL Measurements | Reported Domain | WHOQOL Related Findings |
|---|---|---|---|
| Ratshikana-Moloko 2020 | African Palliative Care Association Palliative care Outcome Scale. | Spirituality/Religion | ^ Engagement in religious and spiritual activities (39.5%). |
| | | Physical Capacity | # Receipt of religious and spiritual care was negatively correlated with pain level. |
| | | Psychological | # Receipt of religious and spiritual care was negatively correlated with feeling that life is worthless. |
| | | Social Relations | # Patients who received religious and spiritual care were likely to have family support. |
| Rayne 2017 | Researcher-developed Self-reported levels of fear | Psychological | # Fearfulness for younger women and who received chemotherapy were higher than older women and who received radiotherapy. |
| Wang'ombe, 2021 | Scale with four dimensions namely level of pain experienced, weight change, quality of sleep and quality of life | Overall QOL | ^ Low overall QOL (56.1%). |
| | | | * Low overall QOL (mean: 24.36). |
| | | Physical Capacity | ^ Low pain (67.1%), weight loss (80.5%), sleeping difficulties (57.3%). |
| | | | * High level of pain (mean: 10.90), weight loss (mean: 4.80), sleeping difficulties (mean: 6.90). |
| Yifru,2021 | MD Anderson Dysphagia Inventory | Overall QOL | * Low overall QOL (mean: 53.34). |
| | | | # Women had a significantly lower QOL; # difficulties in health finance were negatively correlated with overall QOL; patients with initial cancer stages (TI and II) had better QOL; patients with laryngeal/ hypo pharyngeal cancer had significantly lower score in QOL compared with patients with oral cavity/oropharyngeal. |
| | | Physical Capacity | * Low physical wellbeing (mean: 49.44). |
| | | | # Difficulties in health finance was negatively correlated with physical wellbeing; patients undergoing single treatment had worse physical wellbeing; patients with laryngeal/ hypo pharyngeal cancer had significantly worse physical wellbeing compared with patients with oral cavity/oropharyngeal. |
| | | Psychological | * Low emotional wellbeing (mean: 56.63). |
| | | | # Difficulties in health finance was negatively correlated with emotional wellbeing. |
| | | Independence | *Low functional wellbeing (mean: 57.69). |
| | | | # Women had a significantly worse functional wellbeing; difficulties in health finance were negatively correlated with functional wellbeing; patients with lower cancer stages (TI and II) had better functional wellbeing. |

*Note*: all results were for patients with cancer unless indicated otherwise.

*Level (mean)

^prevalence

#Correlates/influencing factors (Note: all as reported by the researchers)

QOL = Quality of life; FACT-G: Functional Assessment of Cancer Therapy-General; FACT-B: Functional Assessment of Cancer Therapy-Breast; EORTC QLQ-C30: European Organization for Research and Treatment of Cancer Quality of Life Questionnaire; EORTC QLQ-BR23: European Organization for Research and Treatment of Cancer Quality of Life Questionnaire-Breast cancer; MSAS-SF: The Memorial Symptom Assessment Schedule Short Form; MVQOLI: Missoula-VITAS Quality of Life Index;GHQ-12: General Health Questionnaire; PTSD: Post traumatic stress disorder.

and being younger (n = 1). Patients with higher levels of emotional intelligence had more internal locus of control orientations (the belief that the reinforcement for the behavior is directly related to the individual's own behavior or qualities), while patients with lower emotional intelligence had more external locus of control orientations (the belief that the reinforcement for behavior is the result of luck, chance or fate or as being under the control of powerful others) (n = 1). Limited access to health information and financial difficulties increased depression and anxiety (n = 2). Religious care reduced the feeling that life was worthless (n = 1). Body image and future perspective were positively related to demographic characteristics such as older age, low educational level, being married, unemployed, and living in urban areas (n = 2). Additionally, compared to older women and those who received radiotherapy, younger women and those who received chemotherapy were more likely to report greater fearfulness (n = 1).

**Level of independence.**   Six studies reported the prevalence of independence-related problems, including difficulties in daily activities (18.9%-75.0%), doing housework (69.2%), mobility (38.7%-75.0%), and their ability to work (26.8%-30.0%). Six studies reported the mean scores of the level of independence. Two studies reported favorable mean scores of levels of independence (e.g., high level of functional wellbeing), whereas four studies reported low functional wellbeing (n = 2), inability to work (n = 1), and difficulties in daily activities (n = 1).

Five studies reported factors influencing the level of independence. Patients' functional wellbeing was positively related to active coping, religious coping, and acceptance (n = 1), but negatively related to difficulties in health finance (n = 1) and caregivers' perceived burden (n = 1). Better functional wellbeing was positively related to demographic characteristics (higher education level, being married, being female) and treatment type (use of combined chemotherapy and radiotherapy, lower cancer stage) (n = 3). Patients with advanced cancer were likely unable to work (n = 1).

**Social relationship domain.**   Seven studies reported how frequently the subjects experienced problems related to the social relationship domain of QOL. Among these studies, 64.3% of the patients reported satisfactory social wellbeing (n = 1) and 29.0%-67.1% reported social and family support (n = 2). In contrast, 26.7%-63.0% of the patients reported reduced social activities (n = 3); 71.0% reported reduced family functioning (n = 1); 50.0%-66.6% reported reduced social support (n = 2); and 27% reported reduced social wellbeing (n = 1). Moreover, 19.0%-56.2% of the patients reported reduced sexual functioning (n = 3), and only 11.2% reported satisfactory sexual functioning (n = 1). One study reported that 38% of caregivers had social support.

Eight studies reported different mean scores of social relationships QOL. Favorable social relationships QOL included high level of social and family wellbeing (n = 2) and moderate to high level of social and family support (n = 2). In contrast, unfavorable social relationships QOL included low level of social wellbeing (n = 2), high social burden of caregiving (n = 1), low social and family support (n = 2), low level of sexual functioning (n = 2), and reduced social activities (n = 1). One study reported high social wellbeing for caregivers.

Ten studies examined factors influencing the social relationships domain of QOL. Participants' higher level of social wellbeing was positively related to being married; having higher level of education; active coping, religious coping, and acceptance; and symptom management (n = 4). In contrast, lower level of social wellbeing was associated with high level of anxiety and health related problems (n = 2). Seeking social support positively influenced family adaptation (n = 1) and reduced depression (n = 1). Having family support was positively related to receipt of religious care and being younger (n = 2). Moreover, having satisfactory sexual functioning was positively related to younger age, being married, and living in urban areas (n = 2).

**Environmental domain.**   Four studies reported the environmental domain of QOL. More than half (52.6%-74.1%) of the patients reported reduction in their income (n = 2); 47.4%-77.9% reported cancer-related extra health expenses (n = 2); 63.0% reported financial difficulties (n = 1); 66.9% reported difficulties in accessing health care (n = 1); and 65.6%-69.0% reported limitation in leisure activities/hobbies (n = 2). One study reported high financial burden among caregivers.

**Spirituality and religious beliefs.**   Seven studies examined the spirituality and religious beliefs domain of QOL. These studies reported that 24.0%-88.9% of the patients engaged in religious and spiritual activities (n = 3), and 55.0% had difficulties concerning the meaning of death (n = 1). Only 5% of caregivers engaged in spiritual and religious activities (n = 1).

Three studies reported the level of spirituality and religious beliefs domain of QOL. Participants reported varying levels of religious and spiritual beliefs (n = 1) but low spiritual wellbeing (n = 1). Only one study reported high spiritual wellbeing among caregivers.

**Table 4. Quality assessment of studies.**

| Criteria | Yes | No | Could not determine |
|---|---|---|---|
| Is the sampling strategy relevant to address the research question? | 27 | 0 | 0 |
| Is the sample representative of the target population? | 15 | 12 | 0 |
| Are the measurements appropriate? | 18 | 4 | 5 |
| Is the risk of nonresponse bias low? | 7 | 3 | 17 |
| Is the statistical analysis appropriate to answer the research question? | 25 | 1 | 1 |

## Risk of bias assessment

Overall, the studies fulfilled the criterion of sampling strategy relevant to address the research question (Table 4). Samples of 15 studies (56%) represented the target population; 18 studies (67%) used appropriate measurements; and 25 studies (93%) used appropriate statistical analysis to answer the research question. However, risk of nonresponse bias could not be determined in 17 studies (63%). A Supplemental file of the results of the quality assessment for each individual study is available in (S2 Appendix).

## Discussion

This review examined the overall and subdomains of QOL and identified their influencing factors among cancer patients and caregivers in SSA. Among the 26 studies that were conducted since 1988 in different SSA countries, among English-speaking patients with various types of cancer, we found significant variations in the sample sizes, the measurements used to assess the overall and subdomains of QOL, and what and how the QOL outcomes were reported. Despite the complexity in QOL research, only three studies were guided by theoretical frameworks. Few studies reported whether or how cultural adaptation was considered when assessing QOL using the measurements developed and used in western countries. There are also inconsistencies and variations in how studies were reported, which complicated comparing findings and drawing conclusions about the QOL and its subdomains. Seventeen of the reviewed papers were published since 2017, indicating increasing attention to cancer-related QOL issues in SSA recently. Most research has been conducted among young female patients with breast and gynecological cancers in South Africa, Nigeria, Kenya, Ghana, Kenya, and Ethiopia. The most common factors that influenced the overall and subdomains of QOL included coping; internal and external locus of control; symptoms (e.g., pain, anxiety, depression, caregiver burden) and symptom management; and religious beliefs and religious care. Moreover, demographics (e.g., age and marital status), cancer-related factors (cancer stage and type of treatment), and social determinants of health (e.g., education, access to information and resources, financial distress, urban vs rural residency) also impacted QOL and its subdomains.

We found that a significant proportion of cancer patients and caregivers in SSA had suboptimal overall QOL and significant health problems related to QOL subdomains. The most common physical health problems included cancer- and treatment-related symptoms, pain, sleeping difficulties, and lack of energy. In the psychological domain of QOL, cancer patients often experienced cognitive function changes (e.g., difficulty in concentration and reduced memory); depression or sadness, anxiety, and PTSD; negative body image; and loss of feelings of pleasure and confidence. In the studies that researched level of independence, more than two thirds of the cancer patients had difficulties in daily activities (e.g., housework, mobility) and inability to work. Regarding the social relationship domain of QOL, patients in a small number of studies reported satisfactory social wellbeing, social and family support, and

satisfactory sexual functioning. In contrast, in most studies patients reported decreased social wellbeing including reduction in social activities, family functioning, social support, and sexual functioning. Of the four studies that examined the environmental domain of QOL, more than half of the participants reported income decrease, extra health expenses due to cancer, reduced leisure activities, and significant financial difficulties and caregiving burden. Finally, in the spiritual and religious beliefs domain, most patients had difficulties with the meaning of death, and many patients engaged in religious and spiritual activities. Patients also had varied levels of religious and spiritual beliefs and low spiritual wellbeing.

Our findings indicate the significant need for recognition and management of QOL-related problems for cancer patients in SSA because they negatively affect different aspects of cancer survivorship (e.g., adherences to cancer therapy) [39]. For example, cognitive impairments affect patients' cancer prognosis, comorbid conditions, or adherences to cancer treatments [40]; psychosocial problems can be barriers to patients' engagement in cancer survivorship care and returning to usual activities in addition to disrupting their adherence to treatment [41]; and patients with worse functional wellbeing often have more difficulties tolerating rigorous cancer treatments and have less favorable treatment outcomes [42]. The prevalence and extent of impacts of these QOL related problems among cancer patients in SSA remain unknown due to the variations in the research methods used, especially study populations, assessment tools, and data analysis methods. Research on the QOL challenges among the increasing number of cancer patients in SSA requires standardized assessment tools and analysis approaches. Despite current limitations in assessment tools that we identified, the African Palliative Care Association African Palliative Outcome Scale (APCA African POS) has proved to offer a potential validity in measuring QOL for African cancer patients receiving palliative care [43]. The APCA African POS has demonstrated sensitivity to change over time assessed in multiple domains among cancer populations [43]. This tool can be used in future research to provide insight into the related QOL domains, which will enable comparison of QOL among cancer populations in different sociocultural settings in SSA [44]. Because all studies reviewed were cross-sectional, it was impossible to know the patterns of change in cancer patients' QOL. Longitudinal research is needed to examine the different QOL challenges during the continuum of survivorship among SSA cancer patients.

Our review findings also highlighted the need for comprehensive supportive care programs in SSA to address QOL issues for cancer patients with limited resources. The QOL challenges patients face in different domains indicated that effective, accessible interventions and clinical care programs should be tailored to patients' needs to improve their QOL. Furthermore, supportive care programs are urgently needed and should include professionals with different expertise (e.g., home care, psychology, occupational health, social worker, and spiritual and pastoral care). Clinicians can assess patients' QOL and related needs and refer them to the providers who can initiate discussions and treatment for related issues (e.g., psychological problems, sexual dysfunction) during clinical care [45, 46]. The clinical care necessitates enhanced medical and allied health education to prepare skillful healthcare professionals with interdisciplinary backgrounds and/or interprofessional collaboration training to take on the oncologic care tasks and meet the complex needs of cancer patients in SSA.

This review also identified the potentially modifiable factors that influenced the overall and subdomains of QOL among cancer patients in SSA. In addition to demographics (younger age, being married, higher education) and medical characteristics (e.g., cancer stage and treatment), better overall QOL and QOL subdomains are related to positive coping; higher internal and lower external locus of control; fewer symptoms (e.g., pain, anxiety, depression, caregiver burden); adequate symptom management; use of religious beliefs and care; and better access to information on cancer and related topics. These demographics and medical characteristics

may be moderators for which QOL prediction models should be stratified, and interventions and supportive care programs can be tailored based on patients' characteristics to maximally improve their QOL [47]. Psychosocial and behavioral factors such as coping, symptom management, use of religious care, and access to information influence cancer patients' stress responses and defense mechanisms and directly or indirectly contribute to a reduction of psychosomatic symptoms (e.g., fatigue, anxiety and depression) and worsening QOL [48, 49]. Consistent with the findings of previous reviews and studies conducted in different parts of the world [48, 50–54], our findings from reviewing these 26 studies indicated that potentially promising interventions against QOL deterioration may include: broadening patients' active coping strategies and acceptance; enhancing religious/spiritual care; managing psychosocial needs, symptoms and functional problems; and improving access to health information. Interventions that use these psychoeducational and psychotherapeutic approaches have improved QOL among cancer patients [55].

This review has certain limitations. We have included studies regardless of their quality because cancer survivorship research in SSA is in its infancy and only a small number of high-quality reports were available. Significant percentages of the reviewed studies did not report appropriate measurements, target population, and nonresponse bias risk information (e.g., nonresponse rate, reasons for nonresponse, and statistical compensation for nonresponse), which may reduce the validity of the study findings. Second, various instruments have been used to assess QOL across studies, making it difficult to draw a definitive conclusion or to precisely interpret the QOL findings. Next, only two studies examined QOL-related issues among caregivers; thus, little is known about the overall and subdomains of QOL among caregivers of cancer patients in the SSA. Another major limitation is that the literature search was restricted to English language publications due to the funding and personnel constraints. Therefore, findings of this review only represent those English-speaking countries in the SSA region, which may have largely limited the cross-cultural generalizability of our findings in French speaking countries in the SSA region. Finally, the reviewed studies mainly focused on younger female cancer patients, which may have limited the generalizability of our review findings.

## Recommendations for future research

1. Describe whether and how the assessment tools are culturally adapted before survey implementation.

2. Conduct research that is theory-based to comprehensively and systematically understand the QOL challenges of local patients and guide intervention development.

3. Conduct research among patients who are diverse in gender and age groups; have different types of cancer; and are from different SSA countries.

4. Conduct longitudinal cohort studies to examine the patterns of change in QOL.

5. Use standardized assessment tools, analyses, and reporting to enable QOL result comparison across studies

6. Establish population norms of the QOL measures among cancer patients in SSA.

7. Conduct research among caregivers to understand their QOL challenges.

8. Develop and test effective, accessible psychoeducational and psychotherapeutic interventions that target potentially modifiable factors to improve the overall and subdomains of QOL for cancer patients and caregivers.

9. Conduct research that targets cancer patients, caregivers, and healthcare providers to guide development of culturally sensitive interventions that are appropriate for the local healthcare resources.

## Recommendations for improving oncologic care in SSA

1. Assess the overall QOL and QOL subdomains among cancer survivors and family caregivers.

2. Prioritize healthcare service resources to optimize QOL for the increasing number of cancer patients.

3. Tailor supportive care programs to better meet patients' and family caregiver needs in different QOL subdomains.

4. Prepare skillful, multidisciplinary healthcare professionals and lay health advisors to take on the tasks of meeting the complex needs of cancer patients.

## Supporting information

**S1 Checklist. PRISMA 2020 checklist.**
(PDF)

**S1 Appendix. Search strategy.**
(DOCX)

**S2 Appendix. Detailed quality assessment.**
(DOCX)

## Acknowledgments

Research support for the study: University of North Carolina-Chapel Hill Lineberger Comprehensive Cancer Center and the School of Nursing.

## Author Contributions

**Conceptualization:** Cloie Dobias, Nilda Peragallo Montano, Ashley Leak Bryant, Lixin Song.

**Data curation:** Yousef Qan'ir, Eno Idiagbonya, Cloie Dobias, Jamie L. Conklin, Chifundo Colleta Zimba, Agatha Bula, Wongani Jumbo, Kondwani Wella, Patrick Mapulanga, Samuel Bingo, Evelyn Chilemba, Jennifer Haley, Nilda Peragallo Montano, Ashley Leak Bryant, Lixin Song.

**Formal analysis:** Yousef Qan'ir, Eno Idiagbonya, Cloie Dobias, Jamie L. Conklin, Chifundo Colleta Zimba, Agatha Bula, Wongani Jumbo, Kondwani Wella, Patrick Mapulanga, Samuel Bingo, Evelyn Chilemba, Jennifer Haley, Ashley Leak Bryant, Lixin Song.

**Investigation:** Yousef Qan'ir, Ting Guan, Cloie Dobias, Patrick Mapulanga, Samuel Bingo, Evelyn Chilemba, Jennifer Haley, Lixin Song.

**Methodology:** Yousef Qan'ir, Ting Guan, Eno Idiagbonya, Cloie Dobias, Jamie L. Conklin, Chifundo Colleta Zimba, Agatha Bula, Wongani Jumbo, Kondwani Wella, Patrick Mapulanga, Samuel Bingo, Evelyn Chilemba, Jennifer Haley, Nilda Peragallo Montano, Ashley Leak Bryant, Lixin Song.

**Project administration:** Yousef Qan'ir, Ting Guan, Ashley Leak Bryant, Lixin Song.

**Resources:** Yousef Qan'ir, Ting Guan.

**Supervision:** Lixin Song.

**Validation:** Yousef Qan'ir, Ting Guan, Lixin Song.

**Visualization:** Yousef Qan'ir, Ting Guan, Lixin Song.

**Writing – original draft:** Yousef Qan'ir, Ting Guan, Lixin Song.

**Writing – review & editing:** Yousef Qan'ir, Ting Guan, Eno Idiagbonya, Cloie Dobias, Jamie L. Conklin, Chifundo Colleta Zimba, Agatha Bula, Wongani Jumbo, Kondwani Wella, Patrick Mapulanga, Samuel Bingo, Evelyn Chilemba, Jennifer Haley, Nilda Peragallo Montano, Ashley Leak Bryant, Lixin Song.

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
