## [Decision Letter · Decision Letter 0]

26 Oct 2021

PGPH-D-21-00606

Quality of life among patients with cancer and their family caregivers in the Sub-Saharan Region: a systematic review of quantitative studies

Dear Dr. Song,

Thank you for submitting your manuscript to PLOS Global Public Health. After careful consideration, we feel that it has merit but does not fully meet PLOS Global Public Health’s publication criteria as it currently stands. Therefore, we invite you to submit a revised version of the manuscript that addresses the points raised during the review process.

EDITOR:

Although, I agree that there are variations/inconsistencies in reporting of QOL relate problems in the published literature and estimation of prevalence and severity of QOL problems is not straight forward. However, looking at the tables and reviewing the findings of published literature, I observed quite a few papers have utilized similar tools to estimate QOL among patients and care givers. So if there is a possibility of making some plots (Forest) for subgroup of studies, which used similar tools and reported scores of same domains (physical, social, spiritual etc.), I would strongly recommend that this type of analysis will strengthen this manuscript as authors are claiming this is a "systematic review of quantitative studies". So rather than just writing that there is variation/heterogeneity/inconsistencies in reported findings of studies, there should be a serious attempt to group some similar studies and run some plots to see effect sizes.

Decision

The peer review process is now complete, my own comments along with both reviewers' comments are attached. Both reviewers have highlighted quite significant points, which need to be addressed in the write-up of this manuscript. Therefore, the final decision can only be made after satisfactory revision of manuscript with due consideration to reviewers'/editor's concerns.

We look forward to receiving your revised manuscript.

Kind regards,

Prof. Kashif Shafique

Academic Editor

Journal Requirements:

1. PLOS Global Public Health has specific requirements for systematic reviews and meta analyses. Please see https://journals.plos.org/globalpublichealth/s/submission-guidelines#loc-systematic-reviews-and-meta-analyses for more information. In order to meet our requirements, please provide:

a) the page numbers on the PRIMA checklist (File S1) where information can be found in your manuscript

b) a Supplemental file of the results of the quality assessment for each individual study assessed, reporting the outcome for each individual criteria considered (this would be more detailed than table 4).

2. Please update the completed 'Competing Interests' statement, including any COIs declared by your co-authors. If you have no competing interests to declare, please state "The authors have declared that no competing interests exist". Otherwise please declare all competing interests beginning with the statement "I have read the journal's policy and the authors of this manuscript have the following competing interests:"

3. Please amend your Data Availability Statement and indicate where the data may be found

4. State what role the funders took in the study. If the funders had no role in your study, please state: “The funders had no role in study design, data collection and analysis, decision to publish, or preparation of the manuscript.”

Additional Editor Comments (if provided):

Reviewers' comments:

Reviewer's Responses to Questions

**Comments to the Author**

1. Does this manuscript meet PLOS Global Public Health’s publication criteria? Is the manuscript technically sound, and do the data support the conclusions? The manuscript must describe methodologically and ethically rigorous research with conclusions that are appropriately drawn based on the data presented.

Reviewer #1: Yes

Reviewer #2: Yes

2. Has the statistical analysis been performed appropriately and rigorously?

Reviewer #1: Yes

Reviewer #2: N/A

3. Have the authors made all data underlying the findings in their manuscript fully available (please refer to the Data Availability Statement at the start of the manuscript PDF file)?

Reviewer #1: Yes

Reviewer #2: Yes

4. Is the manuscript presented in an intelligible fashion and written in standard English?

Reviewer #1: Yes

Reviewer #2: Yes

5. Review Comments to the Author

Reviewer #1: The title is good and reflects the context of the systematic review. The abstract is good but however, the introduction section can be improved upon. The methodology is good. From paper selection down to analysis. However, the inclusion criteria can be set to accommodate papers that were published after the WHO framework on QOL among patients was introduced since that is the model that the study is based upon.

The recommendations are also good, especially research that are theory based which will define the problem phenomenon from the root.

Reviewer #2: Dear authors,

it has been a pleasure to review your manuscript which deals with an important and relevant public health topic in SSA. You display a systematic literature review regarding QoL among cancer patients and their caregivers; a topic that is surely widely neglected so far and need more attention in the future as the burden of cancer diseases in SSA will grow.

The methodology is replicable and understandable for the reader and so are the results. However, I have to 2 points of criticism:

1. The inclusion criteria exclude articles without an English full text. Why is this? This may exclude a massive part of the (West-) African population that is French speaking and used to publish in francophone journals. I assume that some of the "ineligible studies excluded (n=1079)" are excluded because of this. The results show no articles from a French speaking country.

Of course, I do understand the difficulties of having interpreters/translators within the research team and the methodological hurdles using translated texts. Nevertheless, it must be mentioned as a limitation of the study that your result may not represent SSA as a whole.

Before the background of the current discussion around "neocolonialism in Global Health", every researcher needs to be sensitive in the choice of their methodology and must at least discuss around a probable neglecting of local reality through the eyes of "Western researchers" (yes, I acknowledge 3 co-authors from Malawi, however, Malawi belongs to the East-African English speaking region).

2. The second point is that the authors do not discuss the already existing "African APCA POS" outcome measurement tool. The authors mention in their recommendations that a "cultural adapted tool" is needed and future research should have a focus on this. Why not discussing around an existing "African-grown" tool that is available in many local languages. I strongly suggest to adapt the discussion chapter around this tool and acknowledge the already existing "culturally appropiate" efforts that have been made in this field.

Lastly, a minor comment: Lines 302 - 335 are a repetition of the result part. This can be shortened.

Best regards

6. PLOS authors have the option to publish the peer review history of their article (what does this mean?). If published, this will include your full peer review and any attached files.

**Do you want your identity to be public for this peer review?** For information about this choice, including consent withdrawal, please see our Privacy Policy.

Reviewer #1: **Yes: **Mustapha Adebayo

Reviewer #2: No

---

## [Decision Letter · Decision Letter 1]

19 Jan 2022

Quality of life among patients with cancer and their family caregivers in the Sub-Saharan Region: a systematic review of quantitative studies

PGPH-D-21-00606R1

Dear Dr. Song,

We're pleased to inform you that your manuscript has been judged scientifically suitable for publication and will be formally accepted for publication once it meets all outstanding technical requirements.

Within one week, you'll receive an e-mail detailing the required amendments. When these have been addressed, you'll receive a formal acceptance letter and your manuscript will be scheduled for publication.

An invoice for payment will follow shortly after the formal acceptance. To ensure an efficient process, please log into Editorial Manager at https://www.editorialmanager.com/pgph/ click the 'Update My Information' link at the top of the page, and double check that your user information is up-to-date. If you have any billing related questions, please contact our Author Billing department directly at authorbilling@plos.org.

Kind regards,

Prof. Kashif Shafique

Academic Editor

Additional Editor Comments (optional):

Reviewers' comments:

Reviewer's Responses to Questions

**Comments to the Author**

1. If the authors have adequately addressed your comments raised in a previous round of review and you feel that this manuscript is now acceptable for publication, you may indicate that here to bypass the “Comments to the Author” section, enter your conflict of interest statement in the “Confidential to Editor” section, and submit your "Accept" recommendation.

Reviewer #1: All comments have been addressed

Reviewer #2: All comments have been addressed

2. Does this manuscript meet PLOS Global Public Health’s publication criteria? Is the manuscript technically sound, and do the data support the conclusions? The manuscript must describe methodologically and ethically rigorous research with conclusions that are appropriately drawn based on the data presented.

Reviewer #1: Yes

Reviewer #2: Yes

3. Has the statistical analysis been performed appropriately and rigorously?

Reviewer #1: Yes

Reviewer #2: N/A

4. Have the authors made all data underlying the findings in their manuscript fully available (please refer to the Data Availability Statement at the start of the manuscript PDF file)?

Reviewer #1: Yes

Reviewer #2: Yes

5. Is the manuscript presented in an intelligible fashion and written in standard English?

Reviewer #1: Yes

Reviewer #2: Yes

6. Review Comments to the Author

Reviewer #1: The systematic review has been carried out scientifically and fit to be published. The review has highlighted factors that promotes or hinder good quality of life of cancer patients in SSA. These factors are valid as evidently supported in other quantitative studies that were used for the review. The study has also made good recommendations to ensure further culturally friendly approach are used to unveil the problem phenomenon, most especially in SSA.

Reviewer #2: Dear authors, thank you for the revised version of your manuscript. From my side, all questions are answered and changes were made accordingly.

Best regards

7. PLOS authors have the option to publish the peer review history of their article (what does this mean?). If published, this will include your full peer review and any attached files.

**Do you want your identity to be public for this peer review?** For information about this choice, including consent withdrawal, please see our Privacy Policy.

Reviewer #1: **Yes: **Dr Adebayo

Reviewer #2: No
